# Therapeutic Exercise and Conservative Injection Treatment for Early Knee Osteoarthritis in Athletes: A Scoping Review

**DOI:** 10.3390/medicina58010069

**Published:** 2022-01-03

**Authors:** Lucrezia Tognolo, Maria Chiara Maccarone, Stefania De Trane, Anna Scanu, Stefano Masiero, Pietro Fiore

**Affiliations:** 1Department of Neurosciences, Physical Medicine and Rehabilitation School, University of Padua, 35128 Padua, Italy; mariachiara.maccarone93@gmail.com (M.C.M.); anna.scanu@unipd.it (A.S.); stef.masiero@unipd.it (S.M.); 2Neurorehabilitation and Spinal Unit, Institute of Bari, Istituti Clinici Scientifici Maugeri IRCCS, 70124 Bari, Italy; stefania.detrane@icsmaugeri.it (S.D.T.); pietro.fiore@unifg.it (P.F.); 3Rheumatology Unit, Department of Medicine-DIMED, University of Padua, 35128 Padua, Italy; 4Department of Clinical and Experimental Medicine, Physical Medicine and Rehabilitation School, University of Foggia, 71122 Foggia, Italy

**Keywords:** sport, professional athletes, early osteoarthritis, therapeutic exercise, physical activity

## Abstract

*Background and Objectives*: Recent evidence highlighted a higher prevalence of knee osteoarthritis (kOA) among young and former ex-professional athletes. Although the practice of a highly demanding sport is considered a predisposing factor for the knee joint cartilage degeneration, articular cartilage seems to positively respond to a moderate load increase. We aim to investigate recent evidence on the conservative management of early kOA in athletes, with a particular emphasis on therapeutic exercise and injection treatment, in order to highlight whether there are any indications that can influence clinical and rehabilitation practice. *Materials and Methods***:** A scoping review was conducted, screening MEDLINE and PEDro databases for studies published over the past twenty years on the topic. Studies in English, with accessible abstracts, were included in the review. The PICO framework was used (P—patient: athletes, I—Intervention: conservative treatment with therapeutic exercise or injection therapies, C—Comparison: not needed, O—Outcomes: clinical outcomes). Clinical trials, randomized controlled trials, and longitudinal studies were considered. *Results*: Four studies were finally included in the review. Therapeutic exercise seems to have beneficial effects on prevention of cartilage degeneration, on pain reduction, and on physical function enhancement. On the other hand, in mild to moderate stages of kOA the intra-articular viscosupplementation with Hyaluronic Acid showed a medium to long-term improvement in joint pain and function. The Platelet Rich Plasma treatment also showed a significant improvement in pain and function up to 12 months. *Conclusions*: Despite the heterogeneity of the studies considered, a multimodal treatment combining therapeutic exercise and moderate aerobic activity (such as running) should be indicated to prevent kOA development. In cases of symptomatic kOA it may be indicated to add minimally invasive injection therapy that seems to contribute to the improvement of motor function and symptomatology.

## 1. Introduction

Osteoarthritis (OA) is a chronic degenerative joint disease of an inflammatory nature that is characterized by changes in the articular cartilage, the presence of fibrillation area, and cracking and thickening of the subcondral bone [1]. It is considered as the most common form of arthritis [2] and it represents a major cause of disability worldwide, affecting more than 240 million of people and causing symptomatic and activity-limiting concerns [3]. OA is strongly associated with aging and typically affects the knee, hip, spine, and hands, having a considerable impact on costs and mortality rate.

In particular, people affected by hip and knee OA have approximately a 20% higher mortality rate compared with age-matched controls [1]. It has been calculated that in the United States, direct medical costs OA-related exceed 100 billion dollars [4].

Knee osteoarthritis (kOA) is present in 30% of individuals older than 45 years, presenting radiographic evidence of knee OA, of which about a half suffer from OA symptoms [5].

Risk factors for kOA include older age, obesity, and female gender [2,6,7]. However, OA is increasingly being reported in the young and in athletic populations [8]. 

Luyten et al. first proposed the definition and the classification criteria of early OA [9]. This condition is characterized by knee pain, radiographic evidence of Kellgren–Lawrence grade 0 or 1 OA, and at least one of the two following structural criteria: arthroscopic findings of cartilage lesions, MRI findings of articular cartilage degeneration and/or meniscal degeneration, and/or subchondral bone marrow lesions [9]. This is particularly detectable in active and sportive individuals, since signs and symptoms often become manifest only after long periods (and mostly in high-impact loading activities such as running, jumping sports, etc.) [9].

The intensity and the duration of active participation in sport activities seem to be related to the risk of developing OA [10]. Particularly, athletes who practice sports including rapid acceleration with instant deceleration or continuous training with high impact on joints, or those who compete at a professional level for prolonged periods of time, present a greater risk of developing OA [11].

Tran et al., in an extensive systematic review, suggested that the risk of OA may be associated with the type of sport [12]. The occurrence of early kOA, both in amateurs and in professional athletes, has been widely described [13,14,15]. Since previous studies demonstrated that an active lifestyle could protect against the risk of OA development, the Osteoarthritis Systematic International Review and Synthesis Organization (OASIS) stated that the risk of OA is more likely associated with sport trauma rather than with sport activity [10,16]. The anterior cruciate ligament (ACL) and the menisci are the two intra-articular structures more frequently injured following sports trauma and their damage represents a predisposing factor for the risk of OA development [17]. Although post-traumatic osteoarthritis (PTOA) following acute injuries is well recognized, it is assumed that repetitive microtraumas of the joint surface could also be a leading cause of OA, an excessive mechanical stress that can directly damage the articular cartilage and, consequently, negatively alter chondrocytes’ function [18].

The diagnosis of kOA can usually be made by anamnesis and clinical examination revealing knee pain, stiffness, and functional limitation. Radiograph evaluations confirm the physical exam demonstrating the presence of joint space narrowing, marginal osteophytes, subchondral bone sclerosis, and cysts. However, previous studies demonstrated that the damage of the cartilage commonly begins when limited and sporadic symptoms are present, and radiographs alterations are not still detectable [19]. Therefore, since the outcomes may be different depending on the stage of disease evolution, identifying and treating subjects at risk for progression at the early stages could guarantee a better result.

The treatment of kOA requires a multimodal approach including non- pharmacological, pharmacological, and surgical interventions on the basis of disease’s severity and patient’s symptoms [20]. However, most guidelines agree in “core” treatment recommendations for all subjects affected by kOA [21]. Early management requires patient’s education and lifestyle changes with the aim of reducing the mechanical joint load [22,23]. Clinicians are encouraged to provide their patients with necessary information about OA disease progression and self-care techniques, including muscle strengthening, weight management, knee braces for support [24]. Moreover, behavioral intervention represents a useful tool to manage pain, fear, stress, depression, and anxiety in selected individuals [25]. 

Despite poor evidence in the literature, physical therapies are widely used as additional strategies for the management of kOA. Based on the data of a recent systematic review, hich analyzed the effectiveness of physical agents in the management of patients with early OA, transcutaneous electrical nerve stimulation (TENS) and pulsed electromagnetic fields stimulation (PEMFS) demonstrated a pain-relieving action and a beneficial effect on joint function, quadriceps strength, physical performance, and quality of life [26]. Moreover, especially in early-stage OA, functional orthoses could play a role in the management of pain symptom. Both soft-braces and kinesio taping have shown to reduce pain and improves knee joint stability by stimulating cutaneous mechanoceptors and enhancing muscle performance in subjects with kOA [27,28]. 

Guidelines highlight the importance an active lifestyle by limiting sedentary behavior and promoting physical exercise [23,29,30]. However, methods concerning how to achieve this are still unclear.

To the best of our knowledge, there are no comprehensive reviews focused on therapeutic exercise and conservative injection treatment in athletes with early OA. We therefore investigated the most recent research on the conservative management of early kOA in athletes, with a particular emphasis on therapeutic exercise and injection treatment, in order to highlight whether there are any indications that can influence clinical and rehabilitation practice.

## 2. Materials and Methods

### 2.1. Search Strategy

The authors, having clinical and research experience on the topic, formulated a research question defined as follows: What is the most recent evidence for the role of therapeutic exercise or injection therapies in athletes with early OA or predisposing conditions for OA?

The authors followed the procedures of the Preferred Reporting Items for Systematic reviews and Meta-Analysis (PRISMA) 2020 statement.

The PICO framework was used to answer the research question (P, patient, problem, or population: athletes; I, intervention: conservative treatment with therapeutic exercise or injection therapies; C, comparison, control, or comparator: not needed; O, outcomes: clinical outcomes such as motor function, symptom relief, changes in gait analysis, or imaging tests). 

Clinical trials, randomized controlled trials, and longitudinal studies were considered. 

After the research issue was identified, two independent researchers performed a search on the MEDLINE (PubMed) and PEDro database, using the following keywords: “knee osteoarthritis”, “early osteoarthritis”, “athlete”, “sport”, “treatment”, “soccer”, “running”, combined with Boolean operators, looking at OA in the most popular sports, according to Tran’s extensive analysis [12]. A comprehensive process of identifying and selecting appropriate studies was then performed. 

### 2.2. Study Selection

Articles published within the past 20 years, with available abstracts, written in English that answered the question formulated by the authors were included in the review.

Articles that did not report a conservative intervention, those that reported a post-surgical intervention, those that did not describe clinical outcomes, reviews and metanalysis, and articles written in languages other than English were omitted from the analysis.

### 2.3. Data Extraction

Once the articles meeting the criteria for inclusion in the review were chosen, the complete texts were downloaded and examined in depth.

Data were extracted and charted. Authors, year of publication, type of sport, study design, population, sample size, age, intervention and the main results were all extracted and gathered as reported in Table 1.

## 3. Results

### 3.1. Selected Studies

The search yielded 636 studies in total. A total of 95 abstracts were evaluated after an initial screening of titles. Finally, four studies that satisfied the review’s eligibility criteria were chosen. Two of them considered an exercise regimen, while the other two considered injection therapy (Table 1). Figure 1 shows the study selection procedure.

### 3.2. Therapeutic Exercise

Van Ginckel et al., estimated changes in glycosaminoglycan content of knee cartilage measured by gadolinium enhanced magnetic resonance imaging of cartilage (dGEMRIC) imaging, in asymptomatic untrained female novice runners participating in a start to run program (STR), compared to sedentary controls. SRT aimed at jogging for approximately 5 km within a training period of 10 weeks. dGEMRIC index estimates glycosaminoglycan (GAG) content using the anionic contrast agent gadolinium diethylene triamine penta-acetic acid (Gd-DTPA2), that distributes inversely to the fixed negative charge associated with the GAG content.

Before and after the 10-week period, both groups were subjected to dGEMRIC and it was found that the STR contributed to a significant positive change of the median dGEMRIC index compared to sedentary controls (+11.66 ms (95% CI: 25.29, 44.43) vs. 9.56 ms (95% CI: 29.55, 5.83), *p* = 0.006]. Moreover, the change in dGEMRIC index showed a significant improvement with increasing physical activity (*p* = 0.014) [31].

Another study analyzed the effect of an in-field gait retraining program, using mobile biofeedback through a wireless accelerometer and a small wrist mounted running computer [32]. The proposed program reduced cumulative and peak tibiofemoral loads during running in high impact runners with an average age of 251.88 months (20.99 years).

The in-field gait retraining, causing a 7.5% increase in step rate during running, resulted in a significant reduction in tibiofemoral and medial tibiofemoral joint contact forces per stance phase, by the 9.1% and 8.1%, respectively. The reductions in tibiofemoral joint contact forces, noted during the acute gait modification trial, persisted at one-month post-retraining. However, despite the increase in the number of gait cycles needed to cover a given distance, there was no change in cumulative tibiofemoral joint loads [32].

### 3.3. Injection Treatment

In a study evaluating thirty professional male soccer players with a diagnosis of kOA, who underwent two intra-articular injections of a hexadecylamide derivative of hyaluronic acid (HYADD4-G) at one-week interval, and one, three and six months following the treatment, a significant improvement on symptoms, performance of daily activities, local pain, as well as in all of the clinical endpoints was shown. A significant improvement was also found in the knee osteoarthritis and injury outcome score (KOOS) and visual analogue scale (VAS), at one, three, and six months compared to the pre-injection value (*p* < 0.05) [33]. 

In a randomized controlled trial involving professional soccer players with clinical and radiographic (grade I or II Kellgren-Lawrence scale) evidence of degenerative changes in the knee, both Hybrid Hyaluronic Acid (HHA) viscosupplementation and Platelet Rich Plasma (PRP) injections, demonstrated improvement in the clinical outcomes tested at the 3-, 6- and 12-months follow-ups. At the intergroup analysis, HHA group showed better outcomes at the 3- and 6-month follow-ups compared to PRP, losing significance at the 12 months follow-up [34].

## 4. Discussion

Both nonpharmacological and pharmacological treatments are used in the management of OA [30]. Exercise is generally recognized for its benefits in the therapy of OA: it is regarded as an effective, non-pharmacological treatment for improving OA symptoms such as pain and stiffness [24]. Structured land-based exercise regimens, comprising strengthening, aerobic, and balance training, are strongly suggested for the non-surgical care of kOA according to the most recent OARSI guidelines [35]. Mind-body exercises, such as Tai Chi or yoga, are also beneficial and recommended [19].

Moreover, aquatic exercises are recommended for individuals with kOA presenting no other comorbidities, as well as for individuals with cardiovascular or gastrointestinal comorbidities or with pain disorders and/or depression [35].

Regular weight-bearing physical exercise has been shown to have a modulating action in OA molecular pathways through a periarticular trophic effect on bone, muscles, and tendons [36]. Furthermore, chondrocyte biosynthetic activity is susceptible to mechanical stimuli and can influence cartilage morphology and composition [37]. In particular, several experimental research investigating the effect of physical exercise in OA have reported increased synthesis of GAG, anti-inflammatory cytokines, and bone morphogenic proteins, and reduced levels of pro-inflammatory cytokines and MMPs in joint tissues [38]. Therapeutic exercise has also been shown to ameliorate the lubrication of the joints and modulate the production, the viscosity, and the composition of the synovial fluid [39].

Early OA in young and former athletes represents a growing problem with about one third of cases involving the knee joint, particularly considering the rising prevalence of intensive sports practice among the general population [13].

Based on the limited benefits of pharmacologic therapy, and the indications that nonpharmacologic techniques are more effective in relieving symptoms in the long term, preventing or delaying function deterioration, there has been a gradual transition from pharmacologic to nonpharmacologic interventions for kOA treatment [40]. Among non-pharmacological strategies, an individualized therapeutic exercise protocol, personalized according to outcome expectations, pain severity, patient’s preferences, and fear of movement, seems to represent an essential component of the comprehensive and multimodal management of kOA [30]. 

In our review, despite the heterogeneity of the included studies, due to the study design, the interventions used, and the type of sports examined, some important suggestions should be considered. 

A gradually built-up running scheme seems to contribute to a chondroprotective effect of the knee, modulating cartilage GAG content [31]. 

It has been reported that moderate daily physical activity may have a positive effect on cartilage matrix composition, thus contributing to cartilage healing and reducing risk for knee injury [34]. Several studies conducted in animal models of knee PTOA revealed that exercising exerts beneficial effects both at the bone and cartilage level. Treadmill walking has shown to increase bone morphogenic protein expression and to prevent the progression of cartilage and subchondral bone lesions in rats [41]. In addition, aerobic exercise training reported notable results in modulating degenerative process of cartilage and downregulating IL-1β, caspase-3, and MMP-13 expression in ACL transection rat models [42]. In the same experimental model, reduced IL-1β, TNF-α, MMP-13, and caspase-3 expression and enhanced IL-10, IL-4 lubricin, and Hsp70 expression have been detected in articular tissues after moderate physical activity, thus underlying anti-inflammatory, chondroprotective, and anti-apoptotic effects [43,44]. 

Long-distance runners seem to have less musculoskeletal disability, to retain higher functional capacity than age-matched controls, and demonstrated improved cardiovascular fitness [19]. However, a history of long-distance running may cause cartilaginous volume reductions in the tibia, patella, and medial and lateral menisci, possibly as a response to the repeated and continuous load [36]; in ex middle- and long-distance runners a higher incidence of tibiofemoral and patellofemoral OA was found [45].

Therefore, in agreement with the preexisting literature, running activity appears to have a beneficial role on cartilage when appropriately dosed. 

A small increase in step rate may help to minimize peak and impulse contact pressures per step for the tibiofemoral joint while running, thus lowering the incidence of OA. It is noteworthy that the reduction in peak and impulse contact force on medial tibiofemoral compartment is predominant in the medial joint compartment, the most affected during running [32]. Since total tibiofemoral contact force impulse per unit distance is not different between running and walking, cartilage seems to be highly sensitive to peak loads, particularly when applied at a high rate as would be expected in runners with high impact forces [46]. Therefore, a training exercise aimed to reduce peak and impulse tibiofemoral contact forces per stance could represent a relevant tool in preventing early OA in these individuals.

Among the injection treatments, intra-articular (i.a.) viscosupplementation with hyaluronic acid (HA) represents a valid therapeutic strategy for people who regularly practice sport activity and who are diagnosed of mild to moderate kOA [47]. 

HA is a glycosaminoglycan, a natural component of the synovial fluid and the extracellular matrix of articular cartilage, with lubricating, viscoelastic, and “shock absorber” properties. These actions are carried out through the increase of the hyaluronate content in the chondrocytes, the synthesis of proteoglycans, the inhibition of production and activity of pro-inflammatory mediators and MMPs [48,49].

The main international guidelines recommend viscosupplementation with intra-articular (i.a.) HA in the management of kOA as a second-choice conservative treatment in non-responders (or those who have contraindications) to non-steroidal anti-inflammatory drugs. In the case of joint effusion, preventive arthrocentesis has an anti-inflammatory effect thanks to the removal of cytokines, neuropeptides and other inflammatory mediators. This procedure also allows to optimize the therapeutic action of HA by preventing its i.a. dilution [29,30]. 

The two studies included in our review regarding the injection treatment showed a medium to long-term improvement in joint pain and function in a population of young professional football players (age ranges 34–39 and 17–39, respectively) with early kOA. However, the hydrogel solution and the treatment protocols were extremely different.

In the study by Tamburrino et al., patients underwent two i.a. injection of a HYADD4-G hydrogel at one week intervals with no control group [33]. Hexa-decylamide derivative of HA is a highly viscoelastic hydrogel that recovers its original structure even after repetitive mechanical stress. Therefore, it could represent a valid therapeutic option in the integrated treatment of athletes affected by post-traumatic or degenerative early kOA. However, the lack of a control group, and the small sample size, makes the results poorly generalizable.

In the study by Papalia et al., subjects underwent three i.a. knee injections administered at a weekly interval with a HA hybrid formulation (intervention group) or a PRP solution (control group) [50]. In this study clinicians used a hybrid HA solution with a peculiar profile distribution into the joint due to the combination of both high and low molecular weight fractions. These characteristics seem to guarantee good rheological properties with an anabolic activity on chondrocytes, thus stimulating extracellular matrix production [51].

At the 3, 6, and 12-month follow-ups, HHA showed improvement in the clinical outcomes assessed. The PRP group also showed a significant improvement in pain and function recovery up to the 12 month follow up.

Although the exact mechanism of action of the PRP has not yet been clarified, it is hypothesized that there is a stimulating effect on the cell proliferation of chondrocytes, synoviocytes and mesenchymal cells and on the production of cartilage matrix, with an increase in the synthesis of type II collagen and proteoglycans. Furthermore, PRP seems to intervene in the regulation of the chronic inflammatory process by inhibiting the activation of the nuclear transcription factor NF-kB (a regulator of inflammation) and the expression of pro-inflammatory MMPs enzymes, cyclooxygenases 2 and 4, and disintegrins [52].

According to the results obtained from our review, therapeutic exercise seems to represent an appropriate strategy for the management of conditions at risk of developing OA (sports with joint loading or repeated microtraumas). From this perspective, exercise conducted at the right doses, according to a personalized prescription, could not only be employed for rehabilitative programs in patients with kOA at early stages but also for prevention protocols. 

Although different exercise training has been proposed, lower limbs strengthening and general aerobic exercises are recommended by most international guidelines for the treatment of kOA with the main goals of restoring impaired muscle function, resulting in a decreased joint load and a reduced stress on articular cartilage [53]. Previous studies have also demonstrated the role of muscular weakness on the early onset of kOA [54].

Emerging topics include the use of technology supports and remote delivering of therapeutic exercise protocols [55].

An adequate muscle-strengthening rehabilitation program allows a faster recovery of the athlete, prevents the risk of injury recurrence, and slows down the degenerative process of the articular cartilage. Therefore, a strengthening program of knee extensor and hip abductor muscles could also be suggested to prevent kOA development.

A multimodal treatment, based on the identification of subjects at risk, the implementation of preventive interventions with sport-specific programs, and the recourse to minimally invasive interventions in the more severe conditions, could be an appropriate treatment strategy for kOA in athletes. 

In young or former athletes with symptomatic kOA, the combination of viscosupplementation and muscle-strengthening exercises and aerobic training with moderate load (e.g., running with an individualized dose of exercise) appears to be a suitable rehabilitation strategy, since a low-impact training seems not to expose to OA worsening [40].

### Limitations 

This scoping review presents some methodological limitations. In fact, although we deliberately used broad inclusion criteria, in order to avoid excluding relevant articles, the literature research provided only a few studies. Moreover, the selected articles were heterogeneous in study design, study populations, sport considered, and the type of intervention proposed (therapeutic exercise, mini-invasive treatment).

Having conducted the research on a small number of databases may be another limitation of our research. In particular, gray literature was not considered since it is not peer-reviewed and therefore difficult to determine as reliable. 

Excluding case reports from the review may be another limitation. Rehabilitation protocols may have been described in these types of manuscripts and thus were not found by our search.

## 5. Conclusions 

Currently there are no rehabilitation or prevention protocols for athletes with early kOA or conditions at risk of developing it. From the literature review, it seems desirable to combine therapeutic muscle-strengthening exercise and moderate aerobic activity to prevent kOA development. In cases of already developed and symptomatic early kOA it may be necessary to add minimally invasive injection therapy which, from early evidence, seems to contribute to the improvement of motor function and symptomatology. Future studies should be conducted to define specific treatment protocols. 

## Figures and Tables

**Figure 1 medicina-58-00069-f001:**
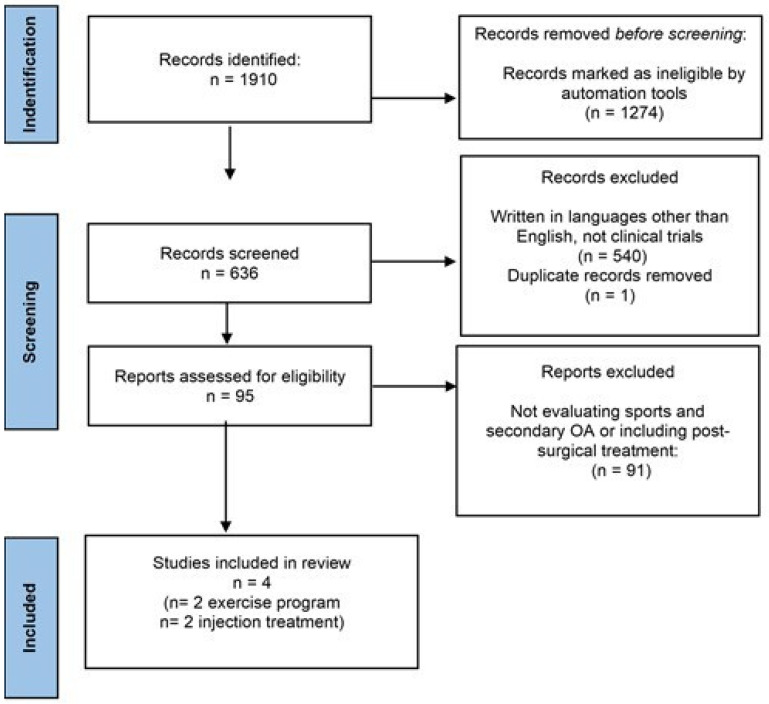
PRISMA Statement flow-chart describing the studies selection strategy.

**Table 1 medicina-58-00069-t001:** Characteristics of the included studies. RCT, randomized controlled trial. i.a., intra-articular.

Authors (Publication Year)	Study Design	N (M/F) Mean (Years)	Sport	Intervention	Outcomes	Evaluation Times	Main Conclusions
Papalia et al., (2016)	RCT	47 (M) 37.2 (range 34–39)	Soccer	3 i.a. injections of HHA (3.2% 64 mg/2 mL, 32 mg High-MW 1100–1400 kDa + 32 mg Low-MW 80–100 kDa) at one week interval or 3 i.a. injections of 5.5 mL PRP	VAS, IKDC, KOOS	Baseline, 3, 6 and 12 months	Both treatments showed to be effective in relieving patients’ symptoms
Tamburrino and Castellacci (2016)	Single arm clinical trial	30 (M) 30.7 (range 17–39)	Soccer	2 i.a. injections of HYADD4-G (3 mL of 8 mg/mL) at one-week interval	VAS, KOOS	Baseline, 1, 3 and 6 months	Significant improvement on symptoms, ADL performance, KOOS and VAS (*p* < 0.05)
Van Ginckel et al., (2010)	Longitudinal	19 (F) 25.5 (range 22–34)	Running	10-week STR program	MRI dGEMRIC index	Baseline and at the end of the program	Significant positive change of the median dGEMRIC index compared to sedentary controls (+11.66 ms (95% CI: 25.29, 44.43) vs. 9.56 ms (95% CI:29.55, 5.83), *p* = 0.006) and with increasing physical activity (*p* = 0.014)
Willy et al., (2016)	RCT	30 (16/14) 20.99 years (range 18–35)	Running	In-filed running retraining program using mobile biofeedback	Derived peak and cumulative tibiofemoral joint contact force estimated by gait analysis	Baseline, at the end of the program, 1 month	7.5% increase in step rate during running with a significant reduction in tibiofemoral and medial tibiofemoral joint contact forces per stance phase (9.1% and 8.1%, respectively)

HHA, hybrid hyaluronic acid; MW, molecular weight; PRP, platelet rich plasma; ADL, activities of daily living; VAS, visual analogue scale; IKCD, international knee documentation committee; KOOS, knee injury and osteoarthritis outcome score; STR, start to run; MRI, magnetic resonance imaging; dGEMRIC, delayed gadolinium enhanced magnetic resonance imaging of cartilage.

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
