# Peer review of "Therapeutic Exercise and Conservative Injection Treatment for Early Knee Osteoarthritis in Athletes: A Scoping Review"

_medicina, 2022, doi:10.3390/medicina58010069_

Round 1
Reviewer 1 Report
Despite revision, editorial mistakes remain to be still critical. It is quite obvious that the authors have not properly checked the submitted version of the manuscript. Many words have been hyphenated awkwardly, which must be fixed by the authors prior to re-submission. In addition, 'reply to reviewers' document is not very reader-friendly, in which most of the responses have been simply 'it has been fixed' or alike without providing detailed amendments and indication of where in the manuscript, such amendments have been made.
Author Response
We thank the Reviewer for his interesting comments that contributed to the improvement of the manuscript.
The text has been thoroughly revised and typing errors have been corrected. Many of the hyphens in the text have been removed. The remaining hyphens represent automatic punctuation marks at the end of the line (e.g. line 44: “com-mon”) and connections between commonly used words composed by two elements.
Refererences have been updated and formatted according to the Journal’s guidelines.
We have also responded to new Reviewers’ comments in detail, highlighting also in the paper the extensive changes made to the text.
Reviewer 2 Report
BRIEF SUMMARY
This review investigated the effects of therapeutic exercise and other conservative treatments for early knee osteoarthritis in athletes. The authors showed a beneficial effect of the therapeutic exercise on the prevention of cartilage degeneration, as well as pain reduction, physical function, and quality of life.
I congratulate the authors on their work. This is a good paper with informative figures and tables. The topic is timely and clinically important. The paper contributes to the clinical decision-making for people at risk of knee OA. However, my main concerns are poor reporting and confusing information flow. Thus, before it can be published, I suggest the authors consider my points below.
SPECIFIC COMMENTS
TITLE
“a literature review”: please clarify what type of review was it; systematic, narrative, or scoping review. In the abstract, you claim it was a scoping review while in the discussion you say it was a narrative review (line 359). Please check the manuscript thoroughly for such examples of contradictions.
“and other conservative treatments’’: this is not what you assessed. Please be specific i.e. you only assessed injections.
ABSTRACT
Please be more specific about the objectives of the review. “….to identify the most recent evidence on the conservative treatment [FOR WHAT?] with a focus on the use of therapeutic exercise in athletes with early kOA”.
Please add conclusions-this is are missing now.
INTRODUCTION
The excessive length and lack of logical information flow in the introduction is off-putting. There is much information that is irrelevant and confuses the reader. Information on diagnosis, biochemistry, treatment, etc can be reduced. Please be sharper in your writing. Currently, it looks like an introduction of a thesis, not a scientific paper.
In the current form, it is quite difficult to figure out from the information flow in the introduction, why it is important to study this, what is the added value of this paper to current knowledge, and who will benefit from this. Please clarify.
METHODS
The whole methods section should be rewritten and structured according to relevant reporting guidelines for literature reviews such as PRISMA. This is too superficial reporting for this to be considered high-quality review. In the current form, it is difficult to judge what exactly was done and how. Just one example: what databases did you search? This is a major concern.
http://www.prisma-statement.org/ . Please report according to PRISMA checklist and include it with your submission as a supplementary file.
Table 1. Please move to the Results section as you present the extracted data. Also please clarify why there is 5 studies included because in the abstract and Figure 1 you mention only 4.?
RESULTS
Line 197: “….a case-control study”: in the abstract you claim you only included clinical trials. Please clarify.
DISCUSSION
Line 252: “… and/or functional orthoses….” . Please provide the below references to support this claim.
https://academic.oup.com/rheumatology/article/57/10/1735/5042121
https://www.sciencedirect.com/science/article/abs/pii/S096663621630073X?via%3Dihub
The discussion lacks a paragraph regarding the limitations of your review. What you present is just the limitations of the studies you included not a limitation of your work.
CONCLUSION
This is just a repetition of the results. Please rephrase and summarize the clinical/practical value of your findings.
Author Response
BRIEF SUMMARY
This review investigated the effects of therapeutic exercise and other conservative treatments for early knee osteoarthritis in athletes. The authors showed a beneficial effect of the therapeutic exercise on the prevention of cartilage degeneration, as well as pain reduction, physical function, and quality of life.
I congratulate the authors on their work. This is a good paper with informative figures and tables. The topic is timely and clinically important. The paper contributes to the clinical decision-making for people at risk of knee OA. However, my main concerns are poor reporting and confusing information flow. Thus, before it can be published, I suggest the authors consider my points below.
Answer: We thank the Reviewer for the interest in our work, and for the useful observations and suggestions.
SPECIFIC COMMENTS
TITLE
“a literature review”: please clarify what type of review was it; systematic, narrative, or scoping review. In the abstract, you claim it was a scoping review while in the discussion you say it was a narrative review (line 359). Please check the manuscript thoroughly for such examples of contradictions.
Answer: We thank the Reviewer for the observation. The study has been designed as a scoping review: we checked the manuscript and corrected the abstract and the study design (please see line 272).
“and other conservative treatments’’: this is not what you assessed. Please be specific i.e. you only assessed injections.
Answer: Thank you for your observation. We have modified the title by inserting the word “injection” to better specify the type of alternative conservative treatment.
ABSTRACT
Please be more specific about the objectives of the review. “….to identify the most recent evidence on the conservative treatment [FOR WHAT?] with a focus on the use of therapeutic exercise in athletes with early kOA”.
Please add conclusions-this is are missing now.
Answer: We modified the Abstract section. The purpose of the study is reported in lines 17 to 19 and the Conclusion paragraph has been added (lines 33 to 37).
INTRODUCTION
The excessive length and lack of logical information flow in the introduction is off-putting. There is much information that is irrelevant and confuses the reader. Information on diagnosis, biochemistry, treatment, etc can be reduced. Please be sharper in your writing. Currently, it looks like an introduction of a thesis, not a scientific paper.
In the current form, it is quite difficult to figure out from the information flow in the introduction, why it is important to study this, what is the added value of this paper to current knowledge, and who will benefit from this. Please clarify.
Answer: we thank the Reviewer for his observations and suggestions. We have reduced the length of this section by removing the useless information for our research purposes (e.g. immunohistological changes in OA). Moreover, we rewrite the Introduction in a single-paragraph section trying to make it more useful, focusing on the topics motivating our scoping review (e.g., the growing incidence of OA in young population, the possible existing link between OA and sport activity, etc.; please see the entire Introduction paragraph). Finally, we added the rationale and the purpose of our research (please see lines 91 to 96).
METHODS
The whole methods section should be rewritten and structured according to relevant reporting guidelines for literature reviews such as PRISMA. This is too superficial reporting for this to be considered high-quality review. In the current form, it is difficult to judge what exactly was done and how. Just one example: what databases did you search? This is a major concern.
Answer: We thank the Reviewer for the useful observations and suggestions. The entire Materials and Methods section has been reviewed and rewritten (please see lines 98 to 130). The search was performed by two independent researcher (L.T. and M.C.M.) on MEDLINE and PEDro databases.
http://www.prisma-statement.org/ . Please report according to PRISMA checklist and include it with your submission as a supplementary file.
Answer: We have uploaded the PRISMA checklist as supplementary file. However, since this is not a systematic review, we are not able to provide all items required by the checklist (identified as N/A Not Applicable in the PRISMA checklist file).
Table 1. Please move to the Results section as you present the extracted data. Also please clarify why there is 5 studies included because in the abstract and Figure 1 you mention only 4.?
Answer: We moved the extracted data to the Results. We can’t find the discrepancy regarding the number of the included studies, since also in the previous version of the manuscript uploaded the authors referred to four studies.
RESULTS
Line 197: “….a case-control study”: in the abstract you claim you only included clinical trials. Please clarify.
Answer: We thank the Reviewer for the observation. The study by Van Ginckel et al. is a longitudinal study, as we reported in the Table 1. We have fixed the error in the text.
DISCUSSION
Line 252: “… and/or functional orthoses….” . Please provide the below references to support this claim.
https://academic.oup.com/rheumatology/article/57/10/1735/5042121
https://www.sciencedirect.com/science/article/abs/pii/S096663621630073X?via%3Dihub
Answer: Due to the excessive length of the paper, the above-mentioned sentence was omitted from the final version of the manuscript, since that sentence lacked information relevant to the purposes of the scoping review.
The discussion lacks a paragraph regarding the limitations of your review. What you present is just the limitations of the studies you included not a limitation of your work.
Answer: A paragraph with the main limits of the study has been added (please see lines 326 to 337).
CONCLUSION
This is just a repetition of the results. Please rephrase and summarize the clinical/practical value of your findings.
Answer: The Conclusion section has been revised to properly clarify the main findings of the literature review as well as the future goals in this field.
Round 2
Reviewer 1 Report
The authors have attended my comments properly.
Author Response
We thank the Reviewer.
Reviewer 2 Report
I thank the Authors for their work.
Regarding:
DISCUSSION
Line 252: “… and/or functional orthoses….” . Please provide the below references to support this claim.
https://academic.oup.com/rheumatology/article/57/10/1735/5042121
https://www.sciencedirect.com/science/article/abs/pii/S096663621630073X?via%3Dihub
Authors Answer: Due to the excessive length of the paper, the above-mentioned sentence was omitted from the final version of the manuscript, since that sentence lacked information relevant to the purposes of the scoping review.
Reviewer comment: The journal does not have a limit on the number of words nor on the number of references. The authors did a good job in reducing unrelated information however I believe the reader would benefit from knowing other established interventions for people with/or at risk of knee OA such as lifestyle-modification (https://pubmed.ncbi.nlm.nih.gov/11567539/ ), exercise (https://www.ncbi.nlm.nih.gov/pmc/articles/PMC3635671/ ), physical modalities ( https://pubmed.ncbi.nlm.nih.gov/25162407/ ) knee orthoses (https://pubmed.ncbi.nlm.nih.gov/29931372/ ) etc. Such crucial information and reference to the suggested work are currently missing in the introduction and should be implemented.
Author Response
We thank the Reviewer for the suggestion. We decided to reduce information in the Introduction section to avoid confounding the reader with useless notions. However, since our paper is focused on therapeutic exercise, we agree with the Reviewer that knowing the benefit of a “core” treatment in patient with OA is crucial. Moreover, we added new references.
This manuscript is a resubmission of an earlier submission. The following is a list of the peer review reports and author responses from that submission.
Round 1
Reviewer 1 Report
Knee osteoarthritis (OA) is a progressive pathological condition prevalent in the senior population but young athletes also suffer increased OA risks. The current review introduces the benefits of exercise intervention on OA prevention and treatment effects. While overall information particularly about biological factors are summarised well, the main weakness is flow of the Results and Discussion. Please see the comments below.
General
- Consider fundamental revision of both Results and Discussion. These two sections are not very distinguishable. Furthermore, each paragraph is rather independent than being interlinked with each other to maintain the flow of the story.
- Please be careful when emphasising positive effects of exercise intervention in the following ways: (1) try not to criticise medication without proper references and (2) try to avoid descriptions such as ‘treatment’, ‘cure’ and ‘therapy’, without references.
- Improve overall readability including grammatical errors.
Specific
- Abstract: ‘Therefore, the aim of the present study it has been to identify…’ Fix the grammatical error.
- Introduction, 2nd paragraph: ‘In particular, people affected by hip and kOA have approximately a 20% higher mortality rate compared with age-matched controls.’ Is this true? If so, please reference this statement.
- Introduction, 3rd paragraph: ‘Risks factors’ Risk factors?
- Introduction, 5th paragraph: ‘Since previous studies demonstrated that sport participation could induce a beneficial effect on knee joint…’ This statement can be misleading. It depends on the type of sports. This part reads as if any kind of sports can give positive effects to the knee joint.
- Introduction, 7th paragraph: ‘They can also subsequently progress causing a relevant joint cartilage impairment.’ This sentence seems to be grammatically incorrect and it is difficult to understand what the authors want to say here.
- Introduction, 8th paragraph: This paragraph summarises OA mechanisms at cellular levels and generally is written well. However, please make it sure that this section fits well in Introduction of the current research.
- Citation: Check the citation styles. For example, is [1-3] required rather than [1][2][3]??
- Introduction, 9th paragraph: ‘Exercise is considered as an effective, nonpharmacological treatment for improving OA symptoms of pain and stiffness’ It sounds like an overstatement if the authors say ‘exercise is an effective treatment for OA symptoms’.
- Results, 1st paragraph: ‘in order to allow a more extensive discuss on the conservative treatment’ Please fix the grammatical error here.
- Results, Table 4: Fundamental improvement is required. Looking at the first example, ‘Gait analysis: no significant improvement was observed from pre-training to post-training in either intervention group. Smaller peak knee flexion angles, extension moments, extensor muscle forces, medial compartment contact forces, and tibiofemoral contact forces were present across group and time, however the magnitude of interlimb differences were generally smaller and likely not meaningful 2 years postoperatively’, this section is difficult to follow. For example, ‘no significant improvement from pre-training to post-training in either intervention group’ is not understandable. What improvement did the authors talk about? What was pre-training and post-training? What were the intervention groups? Not only this first example, but all these summaries in Table 4 are not understandable unless readers actually go through the original papers. Please revise fundamentally to make Table 4 stand alone for readers to understand what they are just by reading Table 4.
- Results, 3.2 Studies on Conservative Treatment: If Table 4 summarises each study, this section should provide overall results.
- Results, 3.3. Studies without intervention: Please refer to the comment 11.
- Discussion, 2nd paragraph: ‘…especially considered the increasing prevalence of…’ Please fix the grammatical error here.
- Discussion, 3rd paragraph: ‘Based on the limited available benefits of pharmacologic therapy… nonpharmacologic techniques are more effective in relieving symptoms in the long term preventing or delaying function deterioration…a gradual transition from pharmacologic to nonpharmacologic interventions.’ Please provide the reference to support this statement. Although exercise interventions have positive effects on OA, there should be some benefits of medication too.
- Discussion, 4th and 5th paragraphs: These two paragraphs do not provide much detailed information. Try to elaborate the sections rather than just saying, ‘… are effective. … are beneficial.’.
- Discussion, 7th and 8th paragraphs: These paragraphs develop good discussion.
- Discussion, 10th paragraph: ‘A modest increase in step rate could contribute in reducing peak and impulse contact forces per step...Cartilage seems to be highly sensitive to peak loads, particularly when applied at a high rate...’ Clarify whether benefits of higher step frequency overweigh increased number of steps.
- Discussion, 11th paragraph: ‘Sports that boost jumping height ability appear to predispose to lower-limb OA’ This sentence does not have strong links with the previous sentence. Does soccer improve a lot of jumping movements?
- Discussion, 12th paragraph: This paragraph is not linked well with the previous section, disturbing the entire flow.
Reviewer 2 Report
The major concern of this study is study design. prevention of cartilage degeneration is the main core of this review. However, the scoping review cannot present this results. The most of outcomes are not correlated the OA findings(ie., MRI data), rather than only functional outcomes (e.g., IDCD, KOOS). In addition, the tile and core of this study is EXERCISE; however, the content is too rough and dispersed ( even including e.g, HA, PRP...etc ). Authors should make focus on the "target".
Round 2
Reviewer 1 Report
The authors have revised the manuscript as per my previous comments. Please attend further comments as below.
- Line 77-78: ‘…that an active lifestyle, thus including level of sport participation…’ What is ‘level of sport participation’?
- Line 289: ‘3.2. Studies on Conservative Treatment’ This entire section lacks proper referencing. Support every statement that is based on the previous research studies.
- Line 291: ‘…looked on estimate the change in…’ I believe this is a grammatical error. There still are quite a few grammatical errors to be corrected. I suggest that the authors responsibly fix all these mistakes. I will not ask for the same thing again.
In general, references should be numbered according to the order of appearance.
Reviewer 2 Report
Thank you for the response. The topic is very interesting and important in sports medicine. Prevention is more important than treatment, especially in OA. However, as your sincere response, this reviewer is still not sure the aim of study clearly. As your reply, the primary aim was to evaluate the current evidence about the role of conservative treatment with therapeutic exercise in treating athletes "with early onset of knee osteoarthritis". However, the review article is not only specific for"early" ​OA. In a word, the inclusive articles are not real early knee OA, since those of papers were not identified. Such a subjective measurement(ie. IKDC and/or KOOS) cannot be diagnosed as "early" knee OA. Above-mentioned is the main concern.